# Enhancing COVID-19 Classification Accuracy with a Hybrid SVM-LR Model

**DOI:** 10.3390/bioengineering10111318

**Published:** 2023-11-15

**Authors:** Noor Ilanie Nordin, Wan Azani Mustafa, Muhamad Safiih Lola, Elissa Nadia Madi, Anton Abdulbasah Kamil, Marah Doly Nasution, Abdul Aziz K. Abdul Hamid, Nurul Hila Zainuddin, Elayaraja Aruchunan, Mohd Tajuddin Abdullah

**Affiliations:** 1Faculty of Ocean Engineering Technology and Informatics, Universiti Malaysia Terengganu, Kuala Nerus 21030, Terengganu, Malaysia or ilanie@uitm.edu.my (N.I.N.); abdulazizkah@umt.edu.my (A.A.K.A.H.); 2Faculty of Computer and Mathematical Sciences, Universiti Teknologi MARA Kelantan, Bukit Ilmu, Machang 18500, Kelantan, Malaysia; 3Faculty of Electrical Engineering & Technology, Pauh Putra Campus, Universiti Malaysia Perlis (UniMAP), Arau 02600, Perlis, Malaysia; 4Centre of Excellence for Advanced Computing, Pauh Putra Campus, Universiti Malaysia Perlis (UniMAP), Arau 02600, Perlis, Malaysia; 5Special Interest Group on Modeling and Data Analytics (SIGMDA), Universiti Malaysia Terengganu, Kuala Nerus 21030, Terengganu, Malaysia; 6Faculty of Informatics and Computing, Universiti Sultan Zainal Abidin (UniSZA), Besut Campus, Besut 22200, Terengganu, Malaysia; elissa@unisza.edu.my; 7Faculty of Economics, Administrative and Social Sciences, Istanbul Gelisim University, Cihangir Mah. Şehit Jandarma Komando Er Hakan Öner Sk. No:1 Avcılar, İstanbul 34310, Turkey; kamil.antonabdulbasah@gmail.com; 8Faculty of Teacher and Education, University Muhammadiyah Sumatera Utara, Jl. Kapten Muchtar Basri No.3, Glugur Darat II, Kec. Medan Tim., Kota Medan 20238, Sumatera Utara, Indonesia; marahdoly@umsu.ac.id; 9Special Interest Group on Applied Informatics and Intelligent Applications (AINIA), Universiti Malaysia Terengganu, Kuala Nerus 21030, Terengganu, Malaysia; 10Mathematics Department, Faculty of Science and Mathematics, Universiti Pendidikan Sultan Idris, Tanjong Malim 53900, Perak Darul Ridzuan, Malaysia; nurulhila@fsmt.upsi.edu.my; 11Department of Decision Science, Faculty of Business and Economics, University Malaya, Kuala Lumpur 50603, Malaysia; elayarajah@um.edu.my; 12Fellow Academy of Sciences Malaysia, Level 20, West Wing Tingkat 20, Menara MATRADE, Jalan Sultan Haji Ahmad Shah, Kuala Lumpur 50480, Malaysia; abdullahmt@gmail.com

**Keywords:** support vector machine, logistic regression, hybrid modeling, small EPV classification, COVID-19 prediction, machine learning classification

## Abstract

Support ector achine (SVM) is a newer machine learning algorithm for classification, while logistic regression (LR) is an older statistical classification method. Despite the numerous studies contrasting SVM and LR, new improvements such as bagging and ensemble have been applied to them since these comparisons were made. This study proposes a new hybrid model based on SVM and LR for predicting small events per variable (EPV). The performance of the hybrid, SVM, and LR models with different EPV values was evaluated using COVID-19 data from December 2019 to May 2020 provided by the WHO. The study found that the hybrid model had better classification performance than SVM and LR in terms of accuracy, mean squared error (MSE), and root mean squared error (RMSE) for different EPV values. This hybrid model is particularly important for medical authorities and practitioners working in the face of future pandemics.

## 1. Introduction

Classification is a technique used to know or estimate a class or a category of an object based on the attributes or characteristics of the object. This approach finds application in numerous fields, encompassing areas such as finance, commerce, healthcare, and industry. Typically, classification serves as a valuable tool for making decisions in situations involving complex problems and extensive datasets. Examples of classification techniques includeaïve Bayes [1,2], decision tree-based approaches [3,4], rule-based methods [5,6], upport vector machines (SVMs) [7,8], neural networks [9,10], k-nearest neighbor (KNN) [11,12], and statistical methods like logistic regression [13,14]. Support vector machine (SVM) and logistic regression (LR) represent two widely utilized supervised classification methods [15,16]. 

Support vector machine (SVM) stands as both a classification and regression technique melding computational algorithms with theoretical underpinnings [16]. These dual qualities have established its strong reputation and fostered its adoption in diverse domains [17]. Typically, the adoption of forecasting techniques hinges on their accuracy and efficiency in handling data. This study finds its primary motivation in the advancement of hybrid forecasting models, with efficiency, accuracy, and precision serving as central themes in prior research and garnering significant attention in various scholarly publications [18,19,20,21,22,23]. 

Since its inception, support vector machine (SVM) has undergone thorough comparisons with various classification methods using real-world data [17,24,25,26,27], yielding valuable insights for scientists. These findings can be summarized as follows: (i) SVM typically demands fewer input variables compared to logistic regression (LR) while achieving the same misclassification rate (MCR) [25]. (ii) In the context of diagnosing malignant tumors from imaging data, SVM and LR exhibited similar MCRs [26,27]. (iii) SVM consistently outperforms LR in classification tasks, as demonstrated in prior research [15]. Beyond these comparisons, it is worth noting that SVM is a parametric method widely employed in machine learning studies. Its recent surge in utilization across various domains, including real-world applications [4,9,13,15,16,28,29], underscores its versatility and effectiveness. However, it is important to recognize that different approaches to training and learning techniques can yield varying levels of prediction accuracy. Therefore, there is a pressing need to investigate and identify the mechanisms that consistently lead to high prediction accuracy when working with SVM models.

In logistic regression analysis, the concept of small events per variable (EPV) can significantly impact the accuracy and precision of regression coefficients associated with independent variables as well as their individual statistical significance tests. EPV is determined by dividing the number of events by the number of predictor variables used in constructing the prediction model [30]. To be more precise, it is the total number of occurrences divided by the total number of variables in the model. When the number of predictors greatly outnumbers the occurrence of outcome events, there is a risk of overestimating or overfitting the model’s predictive performance [31]. As the EPV decreases, the bias in regression coefficients increases, often resulting in extreme values for the maximum likelihood estimate (MLE), which, in turn, affects the accuracy and precision of regression coefficients and their associated statistical significance tests. Research conducted by [31,32,33,34] has pointed out that when the EPV value deviates from the expected minimum values, three types of errors may occur: overfitting (Type I error), underfitting (Type II error), and paradoxical fitting (Type III error). The detection of these errors results in the implementation of multivariable analysis as a general recommendation for the required amount of EPV. According to the research, a particular EPV number is required for confidence in the validity of the model [32,33]. A higher EPV signifies more stable and reliable model estimates while reducing the risk of overfitting. Conversely, a low EPV can result in overfitting and unstable parameter estimates. The Monte Carlo simulation was conducted for small EPV values, i.e., 2, 3, 4, and 5. EPV 2 indicates that for every predictor variable in the model, there are only two events (outcomes of interest). Models with an EPV this low might be at risk of poor generalization to new data. EPV 3, still a relatively low EPV, suggests that there are three events for every predictor variable. While better than EPV 2, it is still a modest value and may warrant caution in terms of model complexity. With EPV 4, there are four events per predictor variable. This is better than EPV 2 or 3 but might still be considered relatively low. Depending on the context, model stability could be improved compared to lower EPV values. EPV 5, a higher EPV value, indicates that there are five events for each predictor variable. While not extremely high, EPV 5 suggests a more balanced relationship between events and predictor variables. This could lead to more reliable model estimates.

In this research, the issue under investigation pertains to the constraints of logistic regression (LR) when it comes to forecasting a small number of events per variable (EPV). This limitation can have an impact on the precision and accuracy of the regression coefficients as well as the statistical significance of their tests. The following is a summary of the study’s hypotheses:

**H1.** *The accurate prediction of the COVID-19 pandemic is important for tracking current and future progress and evaluating countries’ performance related to COVID-19 cases*.

**H2.** *We hypothesize that the hybrid model combining SVM and LR will demonstrate better classification performance compared to SVM or LR alone for predicting small events per variable based on COVID-19 data*.

This new model will integrate the prediction for classification performance and at the same time can improve the accuracy and precision of a small EPV, which is not included in the current model. It is expected that the new hybrid models derived from this study will be able to predict future Coronavirus outbreaks. Modeling COVID-19 with accurate prediction is important for tracking current reductions [20] and future progress [21,22] and evaluating countries’ performance related to COVID-19 cases [23,24]. As far as our understanding goes, there have been scientific investigations related to the COVID-19 pandemic’s spread using a hybrid model that incorporates both logistic regression (LR) and support vector machine (SVM). As a result, this research aims to create a hybrid predictive model that combines LR and SVM to enhance the accuracy of COVID-19 case predictions. The ultimate goal is to provide government authorities and practitioners with valuable insights for effective planning and decision making to curb the global spread of COVID-19.

In order to overcome these obstacles, we developed a prediction mechanism that leverages the advantages of both support vector machine (SVM) and logistic regression (LR). By combining these two methods into a hybrid model, we anticipate achieving more precise results than what can be achieved with either method individually.

## 2. Materials and Methods

### 2.1. Support Vector Machine (SVM)

The novel learning machine known as the support vector machine (SVM) was initially introduced by [17]. SVM offers several advantages that makes it a popular and powerful choice for various machine learning tasks: it is effective in high-dimensional spaces, can handle non-linearity, has relatively fewer hyperparameters compared to some other algorithms, and so on.

Over the years, support vector machine (SVM) has demonstrated remarkable generalization capabilities across various domains, such as bioinformatics [35], text categorization [36], fault diagnosis [37], image detection [38], power systems [39], financial analysis [40], and more. Moreover, [41] highlighted SVM as a valuable approach for making predictions in both classification and regression scenarios. SVM operates by identifying the optimal separator function, or hyperplane, capable of effectively dividing datasets into distinct classes or categories. SVM’s fundamental concept can be explained as follows: input vector x is mapped to a very high-dimension feature space z through some nonlinear mapping, $\varphi \left(x\right),z=\varphi \left(x\right).$ An ideal separating hyperplane is built in this space. For a given training dataset with *n* samples, $\left(x_{1},y_{1}\right),\left(x_{2},y_{2}\right),\ldots \left(x_{n},y_{n}\right)$ where $x_{i}$ is a feature vector in a d–dimensional feature space $R^{d}$ and $y_{i}\in \left\{1,+1\right\}$ is the corresponding class label. The task is to find a classifier with a decision function as shown below:(1)$$f\left(x\right)=w^{T}x+b$$
where w represents the weight vector and b is the bias. The hyperplane is a linear separator that divides space into two parts, which can separate the dataset by maximizing the margins. The best hyperplane is found by maximizing the margin or the distance between two objects from different classes. For non-linearly separable problems, the formula mentioned above can be adapted for use in kernel SVM as
(2)$$f\left(x\right)=\sum_{i}^{N}\alpha_{i}y_{i}K\left(x_{i}x\right)x+\beta_{0}$$

Here, N represents the number of support vectors, where $x_{i}$ is the instant with label $y_{i}$, α is the Lagrange multiplier, $\beta_{o}$ is the bias, and $k\left(x_{i}.x\right)$ is the kernel function. For this study, the simplest kernel function, linear kernel, was used, which is expressed as:(3)$$k\left(x,y\right)=x.y+c$$

### 2.2. Logistic Regression (LR)

LR is one of the most commonly applied classification methods in medical data analysis. Despite its popularity, this methodology can produce inaccurate estimates of class membership [25,42], and the difficulty increases when working with a limited sample set [31,32,33,34]. The LR model following [31,32,43] can be written as:$$y\left(x\right)=\frac{e^{\beta_{0}+\beta_{1}x_{1}+\ldots +\beta_{j}x_{p}+\varepsilon_{i}}}{1+e^{\beta_{0}+\beta_{1}x_{1}+\ldots +\beta_{j}x_{p}+\varepsilon_{i}}}$$
where $y\left(x\right)$ is the probability of success with the probability $0\leq y\left(x\right)\leq 1$ and $\beta_{j}$ is the parameter with j=1,2,…,p.

Within this model, the x‘s represent the covariates used for classifying the response, and the *β_i_* variables stand for the regression coefficients. The logit, represented as log(1/1 − π), signifies the odds ratio of classifying the response into category one rather than zero. When employing logistic regression, predicting the class involves computing probabilities. In essence, the logistic model (also known as the logit model) in statistics is a statistical framework where the likelihood of an event occurring is modeled by expressing the log-odds for that event as a linear combination of one or more independent variables. Classifying binary responses based on logistic regression analysis involves utilizing the probability model with the following criteria: If the probability yielded from the model is less than 0.5, then the predicted result is category zero. If the probability of the model is greater than or equal to 0.5, then the predicted result is category one.

### 2.3. Proposed Hybrid Model

The importance of prediction models lies in their ability to enhance decision making through increased accuracy, efficiency, and precision. This significance underscores why these models remain essential, in-demand, and dynamic. However, these essential attributes are notably absent in support vector machine (SVM) and logistic regression (LR) models. SVM and LR models have indeed proven effective within their respective linear and nonlinear domains, but it is important to acknowledge that they do not offer a universally applicable solution for all scenarios. As a result, a hybrid approach that leverages both linear and nonlinear modeling capabilities is suggested as a means to enhance overall prediction effectiveness. Consequently, there is a dearth of research focusing on improving predictive model effectiveness, particularly in the context of COVID-19 in Malaysia, where SVM and LR models are concerned. This study advocates for the adoption of hybrid models for two primary reasons. Firstly, relying solely on individual SVM and LR models may not suffice for capturing all predictive characteristics. Secondly, one or both of these models may fail to recognize the actual data generation process.

### 2.4. Statistical Performance Criteria

As previously mentioned, previous studies such as [12,31] encountered issues related to overestimating EPV due to their constrained sample sizes. To investigate this matter, a series of simulations with low EPV values (specifically, 2, 3, 4, and 5) was generated following the approaches outlined in [31,32]. To evaluate the comprehensive performance of the proposed hybrid models, we employed established statistical criteria, as recommended by [44,45], which included accuracy, MSE (mean squared error), and RMSE (root mean squared error):(4)Accuracy=TP+TN/TP=TN+FP+FN
(5)$$\mathrm{RMSE}=\sqrt{\frac{1}{n}\sum_{i=1}^{n}{\left(y_{i}-\frac{1}{n}{\hat{y}}_{i}\right)}^{2}}$$
(6)$$\mathrm{MSE}=\frac{1}{n}\sum_{i=1}^{n}{\left({\hat{y}}_{i}-y_{i}\right)}^{2}$$

### 2.5. Data

Machine learning has been applied to COVID-19 research [46,47,48]. In this paper, we used a novel Coronavirus dataset from Dec 2019 to May 2020 provided by the WHO Coronavirus Disease database to assess the performance of hybrid, SVM, and LR models with different events per variable (EPV 2 to EPV 5). There are four characteristics in this dataset, which are Y = pprobability of deaths, $x_{1}$ = state/province, $x_{2}$ = confirmed cases, and $x_{3}$ = recovered cases. In total, 3,568,217 samples of different ages and genders were used in this study. Out of these samples, 1,157,370 cases were recovered. The dataset included information on 248,347 deaths that occurred at 210 different sites worldwide. A higher number of recoveries might indicate that milder cases are prevalent in a province, leading to a lower probability of death. A high number of confirmed cases and a low number of recoveries, on the other hand, may indicate that the disease is fatal [49,50,51,52]. We performed data analysis using the Python programming language. The analysis involved utilizing various Python libraries, including NumPy for numerical operations, Pandas for data manipulation, and Matplotlib and Seaborn for data visualization. The flow process of the hybrid model is summarized in Figure 1 below. In our initial step, we applied SVM classification to our training dataset. For this study, we allocated 70% of the data for training and reserved 30% for testing, which is a common and reasonable split in the field of machine learning. Subsequently, we assessed the accuracy of this classification using Equation (4). Likewise, we applied LR to the same training dataset, performed classification, conducted testing, and calculated its accuracy using the same equation. Finally, we moved on to predict the combined output from both SVM and LR, which constituted our hybrid model. In this approach, we considered the classifier that yielded a superior result for each training datapoint. Specifically, when SVM provided a better outcome, we opted for SVM, and conversely, when LR offered a superior output, we selected the LR output. Situations where both classifiers yielded identical results were considered optimal. Now, leveraging this trained data, we proceeded to predict the output for the testing dataset. Once again, we determined the accuracy of these predictions using Equation (4). Through this process, we carefully observed and established that the hybrid model consistently delivered superior accuracy and innovation in forecasting future Coronavirus outbreaks.

## 3. Results and Discussion

ANOVA was used to analyze the logistic regression model in order to test the difference between more than two means. Table 1 shows that the logistic model was indicated to be statistically significant since the *p*-value for chi-square was 0.000 and less than the significance level (0.05).

This study primarily emphasizes achieving the best balance of consistency and efficiency, particularly when dealing with small EPV values, among the hybrid, LR, and SVM models. 

Figure 2 and Figure 3 show the comparisons of coefficient values for the second variable at different numbers of events per variable (from EPV 2 to EPV 5) between LR and SVM, respectively. For example, by looking at EPV 2 for the second variable, the mean values for LR from EPV 2 to EPV 5 were 0.13, 0.13, 0.16, and 0.16, while for SVM, the mean values from EPV 2 to EPV 5 were 0, 0, 0.03, and 0.03. This suggests that as the EPV values decrease, the frequency distribution of estimated regression coefficients tends to concentrate more towards a normal distribution with an approximate mean of zero [53]. In simpler terms, when EPV decreases, the distribution becomes “flatter,” especially in the case of the LR distribution, while the SVM distribution becomes less peaked and exhibits thinner tails. The standard deviation values for LR from EPV 2 to EPV 5 were 0.349, 0.349, 0.305, and 0.305. For SVM, the standard deviation values from EPV 2 to EPV 5 were 0.13, 0.13, 0.121, and 0.121.

Figure 4 and Figure 5 show the comparisons of coefficient values for the third variable between LR and SVM, respectively. By looking at EPV 2 for the third variable, the mean values for LR from EPV 2 to EPV 5 were 0.14, 0.12, 0.12, and 0.12. For SVM, the mean values from EPV 2 to EPV 5 were −0.1, 0.01, 0.05, and 0.05. The standard deviation values for LR from EPV 2 to EPV 5 were 0.287, 0.356, 0.322, and 0.322. For SVM, the coefficient values from EPV 2 to EPV 5 were 0.126, 0.166, 0.181, and 0.181. These findings offer evidence that the stability of coefficient values is higher for SVM than for LR in the case of both the second and third variables. As mentioned by [32,33], small EPV values lead to inconsistent coefficients. Therefore, it is important to propose a new class of hybrid model based on LR and SVM for predicting small EPVs. This new model will integrate the prediction for classification performance and at the same time can improve the accuracy and precision of small EPVs, which is not covered in the current model.

Figure 6 and Figure 7 show the comparisons of coefficient values for the second and third variables for the hybrid model. The mean values for the second variable from EPV 2 to EPV 5 were −0.0360, −0.0219, −0.1489, and 0.1464, while for the third variable, the mean values were −0.0283, −0.0229, 0.1015, and 0.1020. The standard deviation values from EPV 2 to EPV 5 were 0.1979, 0.2247, 0.2741, and 0.2717 for the second variable, while for the third variable, the coefficient values from EPV 2 to EPV 5 were 0.1617, 0.2045, 0.3090, and 0.3086. As illustrated in the above figures, it is clear that the values of the mean and standard deviation for the hybrid, LR, and SVM models were unstable as EPV values increased. Through demonstration of improved estimating performance at larger scales, the proposed model aims to predict accuracy, MSE, and RMSE as in Equation (4), Equation (5), and Equation (6), respectively. 

The proposed model’s effectiveness was further examined by focusing on low EPV cases. Values for accuracy, MSE, and RMSE are shown in Table 2, which presents a comparison of three accuracy indices across various EPV values. The best-performing parameter sets were chosen based on their high accuracy and low MSE and RMSE values. Accuracy, MSE, and RMSE values for the hybrid model were comparatively lower than the corresponding values for the LR and SVM models.

Accuracy, MSE, and RMSE values were converted to diagrams and are displayed in Figure 8. All coefficients were found to be unstable as EPV increased. However, when we look at the accuracy values, the hybrid model is more powerful with high accuracy compared to LR and SVM for EPV 2 to EPV 5 values. 

This paper examines a comparison between the LR and SVM models for predicting COVID-19 cases for different numbers of events from EPV 2 to EPV 5. The results show that the hybrid model with EPV 5 is capable of obtaining good generalization prediction accuracy, MSE, and RMSE compared to LR and SVM. Small EPV values can cause inconsistent coefficients, as pointed out by Peduzzi [32,32]. Therefore, it is important to propose a new hybrid algorithm to obtain optimal parameters and improve the prediction accuracy, MSE, and RMSE.

The main highlight of this study is the evaluation of a new hybrid model that combines support vector machine (SVM) and logistic regression (LR) for predicting small events per variable (EPV) in the context of COVID-19 classification. This study demonstrates that this hybrid model outperforms SVM and LR alone in terms of accuracy, mean squared error (MSE), and root mean squared error (RMSE) for different EPV values. This finding is particularly significant for local medical authorities and practitioners who are involved in managing and responding to future pandemics. This study addresses the limitation of LR in predicting small EPVs, which can impact the accuracy and precision of regression coefficients. The proposed hybrid model provides a more accurate and efficient approach for COVID-19 prediction, contributing to the field of machine learning classification in the context of a previous infectious disease that provides a preemptive basis for future pandemic research.

## 4. Conclusions

The use of this proposed model is anticipated to yield advantageous outcomes for the community. The improved predictive capabilities of the model will prove valuable to the government, particularly the Ministry of Health, in formulating strategies and identifying necessary measures to enhance the management of COVID-19 cases. Additionally, it will benefit the general public by helping them comprehend and take preventative measures to stop the spread of novel Coronavirus illnesses.

This study acknowledges that support vector machine with linear kernels is a quick technique, but it may not always provide the most accurate classifications compared to support vector machine with nonlinear kernels. The distribution of the training process for support vector machine with nonlinear kernels presents challenges, and as a result, these methods were not utilized in the present study.

For future research, it would be valuable to explore the application of support vector machine with nonlinear kernels to further improve the classification accuracy. Investigating other ensemble methods or hybrid models that combine SVM and LR with different machine learning techniques could provide even more accurate and precise predictions for COVID-19 cases or future pandemics. Considering a broader range of data sources and longer time frames could enhance the robustness and reliability of the predictive models. Additionally, using different machine learning methods, such as k-nearest neighbor (k-NN), which is catered to the classification of COVID-19 [54], would also have a positive impact on future studies. Conducting real-world validation and testing of the proposed hybrid model in different geographical regions or different ecological settings and for other infectious diseases would validate its effectiveness, applicability, and reproducibility in various conditions and contexts.

## 5. Limitation of the Study

Although SVM trained with linear kernels is quick to train, it does not always outperform SVM trained with nonlinear kernels in terms of classification accuracy. Distributing the support vector machine training process using non-linear kernels is challenging, so this approach was not implemented in this work.

## Figures and Tables

**Figure 1 bioengineering-10-01318-f001:**
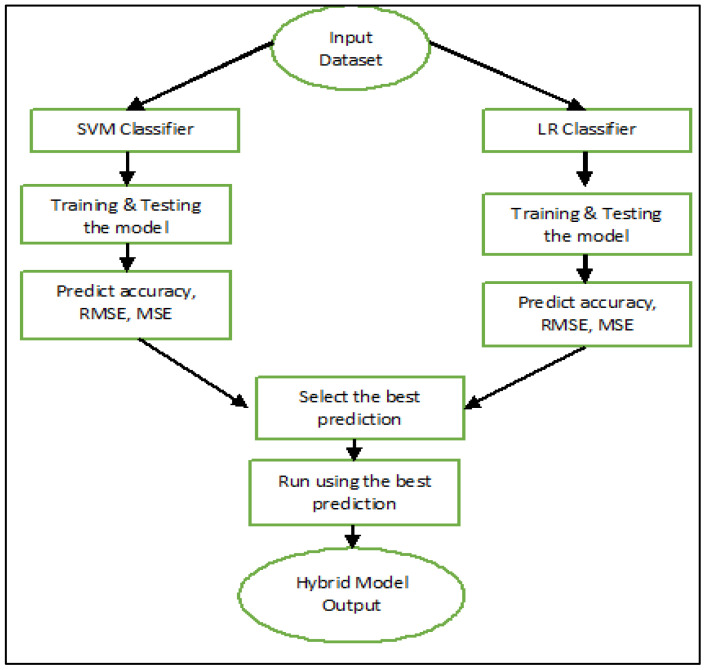
Flow process of hybrid novel Coronavirus dataset.

**Figure 2 bioengineering-10-01318-f002:**
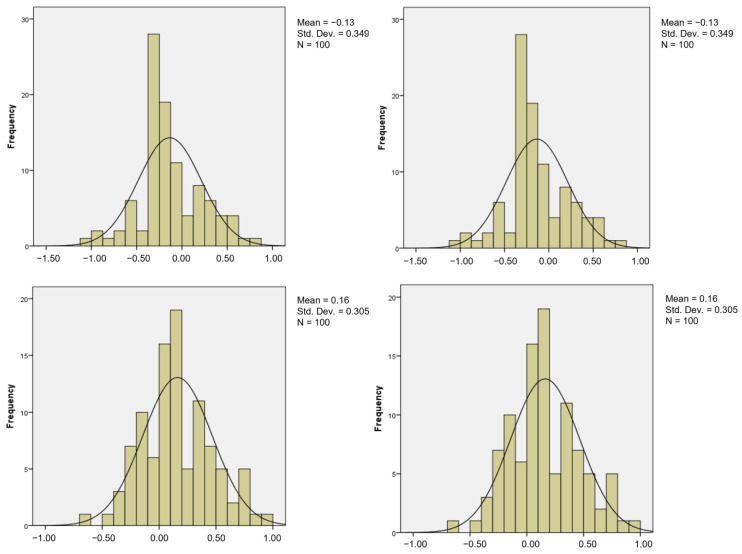
Number of events per variable and frequency distribution of estimated regression for LR (second variable).

**Figure 3 bioengineering-10-01318-f003:**
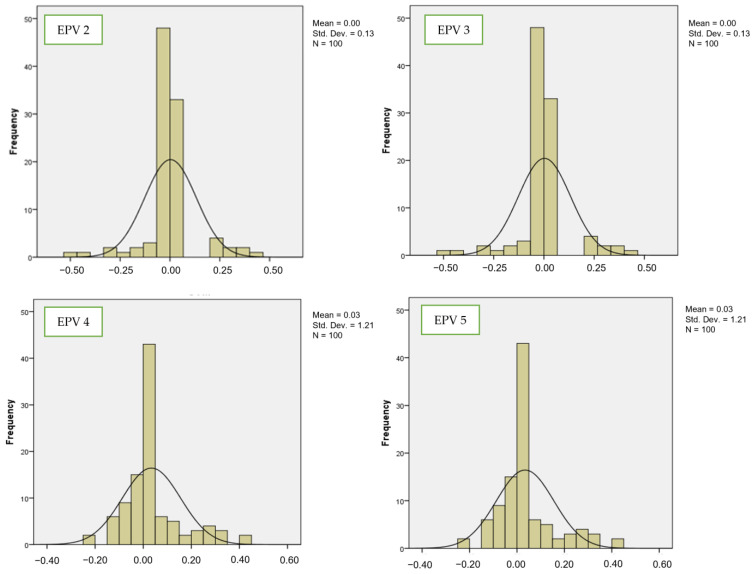
Number of events per variable and frequency distribution of estimated regression for SVM (second variable).

**Figure 4 bioengineering-10-01318-f004:**
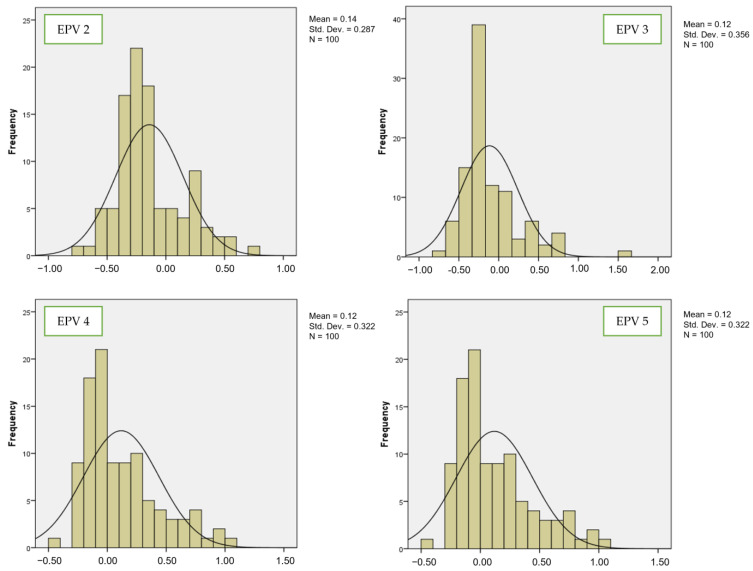
Number of events per variable and frequency distribution of estimated regression for LR (third variable).

**Figure 5 bioengineering-10-01318-f005:**
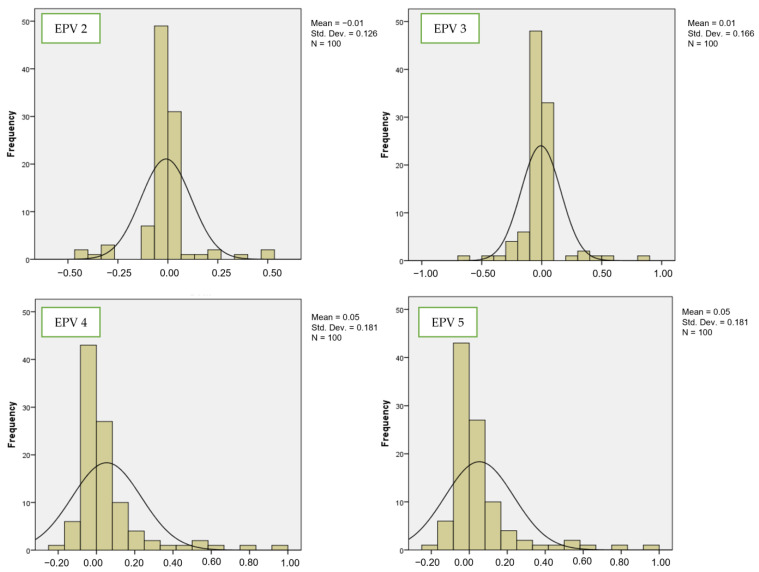
Number of events per variable and frequency distribution of estimated regression for SVM (third variable).

**Figure 6 bioengineering-10-01318-f006:**
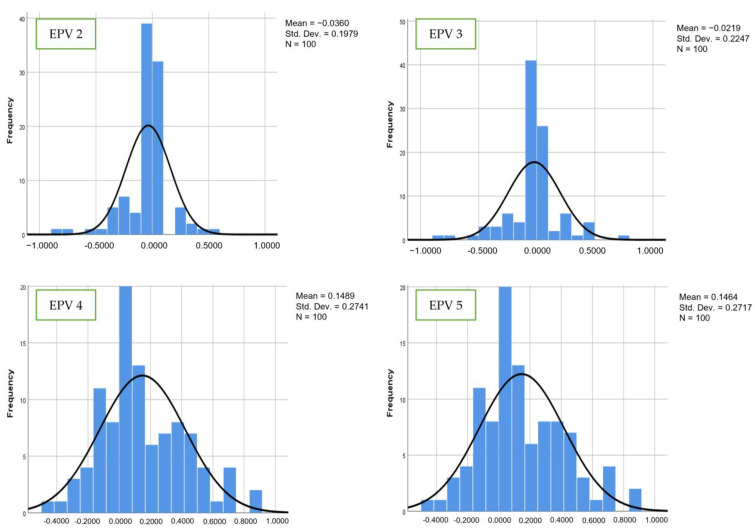
Number of events per variable and frequency distribution of estimated regression for hybrid model (second variable).

**Figure 7 bioengineering-10-01318-f007:**
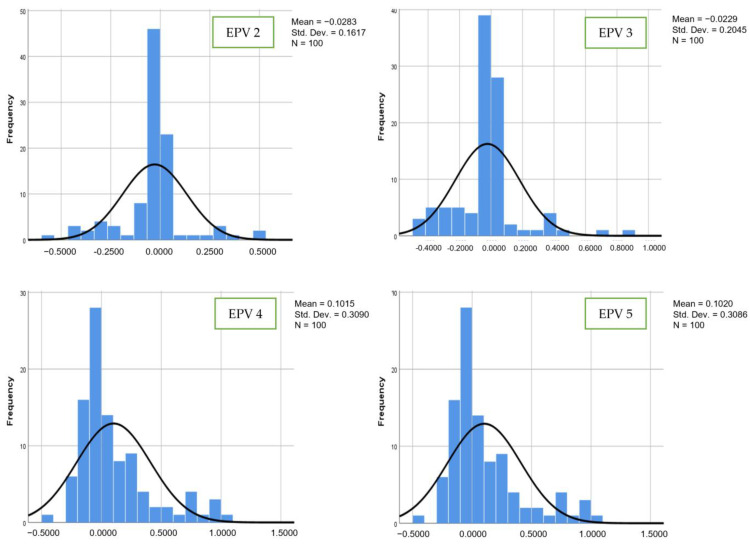
Number of events per variable and frequency distribution of estimated regression for hybrid model (third variable).

**Figure 8 bioengineering-10-01318-f008:**
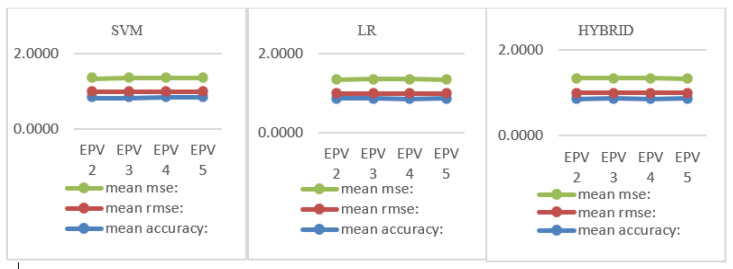
Number of events per variable, MSE, and RMSE for LR and SVM.

**Table 1 bioengineering-10-01318-t001:** Model summary for logistic regression.

Chi-Square	Significant *p*-Value	Interpretation
150.627	0.000	model significant

**Table 2 bioengineering-10-01318-t002:** Summary of performance results obtained with the accuracy, MSE, and RMSE values for LR, SVM, and hybrid models.

	EPV 2			EPV 3		
	LR	SVM	Hybrid	LR	SVM	Hybrid
Accuracy	0.8649	0.8538	0.8684	0.8642	0.8538	0.8691
RMSE	0.1351	0.1462	0.1316	0.1358	0.1462	0.1309
MSE	0.3539	0.3660	0.3485	0.3558	0.3668	0.3485
	EPV 4			EPV 5		
	LR	SVM	Hybrid	LR	SVM	Hybrid
Accuracy	0.8622	0.8540	0.8684	0.8678	0.8549	0.8715
RMSE	0.1378	0.1460	0.1316	0.1322	0.1451	0.1285
MSE	0.3596	0.3670	0.3496	0.3523	0.3665	0.3469

## Data Availability

Not applicable.

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
