# Peer review of "Enhancing COVID-19 Classification Accuracy with a Hybrid SVM-LR Model"

_bioengineering, 2023, doi:10.3390/bioengineering10111318_

Round 1
Reviewer 1 Report (Previous Reviewer 1)
The authors have provided rational responses to my issues. Hence, this work can be accepted in its current state.
Author Response
We successful address all the reviewer comment

Reviewer 2 Report (New Reviewer)
I want to thank you for the opportunity to review the manuscript. This is an interesting topic. However, the paper presented requires some modifications to make it more straightforward and organized.
The introduction is confusing and disorganized. At times, I thought I was reading an introduction. Suddenly, some hypotheses are presented. Afterward, the text resembles a description of the method, and finally, the authors conclude the study.
- H1 doesn't seem like a hypothesis to me. In fact, it is an argument to justify the study or its relevance.
-H2: The authors do not present any prior theoretical argument that justifies why they expect the hybrid model to be better than the others. This sounds like a hypothesis written after the study was done.
The authors begin the methods session with information that resembles a theoretical review. Important and detailed information about the database (variables, number of subjects, outcomes...) was presented as unsatisfactory. Information about the procedures, software, percentage of data used to train the algorithms, sample size used in tests, and statistical tests were also presented confusingly and incompletely.
The discussion of the results is superficial, and the study's conclusions exceeded the previously described objectives. I suggest that in the conclusion, the authors limit themselves to answering the purposes. Some elements present in the conclusion should be included to discuss the findings.
Author Response
We successful address all the reviewer comment

Reviewer 3 Report (New Reviewer)
I have read the paper. My over all opinion is that it is a good paper with a plausibile approach (SVM + logistic regression) to the modeling of some of the various issues related to covid-19. The results are convincing and line with others I have read.
Nonetheless, I have a concern, or better an issue, about the method. While I appreciate and approve that a ML method has been used, instead of resorting to DL(deep learning) that in this case I would not find appropriate, nonetheless there are a lot of methods in the ML domain (beyond SVM) that could have obtained similar or even better results.
At this point I do not ask the authors to try to campare SVM with other ML algorithms. Nonetheless this issue should be mentioned for example in the Intro and in the final discussion citing an article where several ML methods were compared to find an optimum for a similar problem ion the covid-19 field. The paper is the following:
- aa vv., Is a COVID-19 second wave possible in Emilia-Romagna (Italy)? Forecasting a future outbreak with particulate pollution and machine learning, Computation, Volume 8, Issue 3, 2020, doi: 10.3390/computation8030074
With this final modification the paper would become of acceptable quality.
Author Response
We successful address all the reviewer comment

Round 2
Reviewer 2 Report (New Reviewer)
All the points highlighted in the previous review were adequately answered and adjusted.
Reviewer 3 Report (New Reviewer)
The paper is now of publishable quality
This manuscript is a resubmission of an earlier submission. The following is a list of the peer review reports and author responses from that submission.
Round 1
Reviewer 1 Report
Support Vector Machine and Logistic Regression are two different statistical classification methods and they play significant role in statistical analyses of data. The authors propose a new hybrid model based on Support Vector Machine and Logistic Regression for predicting small Event per Variable. Their work suggests that their hybrid model shows better classification performance than Support Vector Machine and Logistic Regression in terms of accuracy. This works is medical authorities and practitioners working in the face of future pandemics. Thus, I recommend that this work can be accepted after minor revision.
1. The authors should improve the resolution of Figure 2-5 so that these two figures are clearly identified.
2. The authors provide more details of machine learning used to study COVID-19.
Author Response
All comment was address carefully.

Reviewer 2 Report
Although the research topic is important, this is a very confusing article
1. The research topic says that the accuracy of the classification is to be done, but what kind of classification is to be done is not well described in the article
2. SVM has been around for a long time, so it is a bit too much to say that it is a newer machine-learning algorithm
3. In the data and description in the second section, the output variable is the probability of death, so the topic of this article is to predict the probability of death? The input variables include provinces, the number of confirmed cases, and the number of recoveries. Such input and output variables make people question why the probability of death is related to the input variables, which requires a theoretical basis to support
4. What exactly is the definition of EPV 2 to EPV 5? It needs a detailed explanation
5. The flow chart in Figure 1 is not considered an innovation. How to choose the best prediction is not explained in detail in the article, so how to combine the two methods of SVM and LR into a Hybrid method still needs to be detailed illustrated.
6. From the research results in the third section, it is difficult to see how to predict the probability of death. The comparison of the three methods still requires a statistical test of Anova or Manova.
7. Although the author emphasizes an innovative method, there is no innovation in the article, so the author still needs to explain clearly
Author Response
All comment was address carefully.
